# Spatial Distribution of, and Variations in, Cold Regions in China from 1961 to 2019

**Yumeng Wang** [1], **Jingyan Ma** [2,*], **Lijuan Zhang** [1], **Yutao Huang** [1], **Xihui Guo** [1], **Yiping Yang** [1], **Enbo Zhao** [1], **Yufeng Zhao** [1], **Yue Chu** [1], **Meiyi Jiang** [1] and **Nan Wang** [1]

[1] Heilongjiang Province Key Laboratory of Geographical Environment Monitoring and Spatial Information Service in Cold Regions, Harbin Normal University, Harbin 150025, China; wangyumeng19980603@163.com (Y.W.); zlj19650205@163.com (L.Z.); huangyutao0128@163.com (Y.H.); 18235889220@163.com (X.G.); ep_yangyiping@163.com (Y.Y.); zeb030410@163.com (E.Z.); janekabesilsen@163.com (Y.Z.); cy254654@163.com (Y.C.); jiangmi199608@163.com (M.J.); wangnan199710@163.com (N.W.)

[2] Teaching Management and Teaching Office, Harbin Normal University, Harbin 150025, China

[*] Correspondence: 18686818800@163.com

**Abstract:** In this study, on the basis of the temperature data collected at 612 meteorological stations in China from 1961 to 2019, cold regions were defined using three indicators: an average temperature of $< -3.0\,°C$ during the coldest month; less than five months with an average temperature of $>10\,°C$; and an annual average temperature of $\leq 5\,°C$. Spatial interpolation, spatial superposition, a trend analysis, and a spatial similarity analysis were used to obtain the spatial distribution of the cold regions in China from 1961 to 2019. Then, the areas of the cold regions and the spatial change characteristics were analyzed. The results reveal that the average area of the cold regions in China from 1961 to 2019 was about $427.70 \times 10^4\,km^2$, accounting for about 44.5% of the total land area. The rate of change of the area of the cold regions from 1961 to 2019 was $-14.272 \times 10^4\,km^2/10\,a$, exhibiting a very significant decreasing trend. On the basis of the changes between 1991–2019 and 1961–1990, the area of China's cold regions decreased by $49.32 \times 10^4\,km^2$. The findings of this study provide references for studying changes in the natural environment due to climate change, as well as for studying changes on a global scale.

**Keywords:** cold region of China; spatiotemporal distribution; spatiotemporal variation; 1961–2019

## 1. Introduction

Cold regions are a valuable part of the earth system [1], and they are generally defined as areas with low temperatures [2] and with the presence of ice and snow for at least part of the year. Therefore, the ice and snow resources in cold regions are very rich and are very important to human production activities [3]. They are an essential and lasting factor that affects the natural and social systems on the Earth's surface [4]. Cold regions are very sensitive to climate change [5], and climate change has changed the distribution of the cold regions around the world [6–10]. The sixth report of the IPCC pointed out that the global annual average surface temperature has risen by 1.09 °C in the past 100 years, that the climate warming trend in China was much higher than the global average [11,12], and that the area and spatiotemporal distribution characteristics of China's cold regions are undergoing significant changes. However, at present, little attention has been paid to the temporal and spatial changes in the cold regions in China. Gaining an accurate understanding of the distribution of, and variations in, cold regions has important theoretical and practical significance for engineering projects in cold regions, industrial and agricultural production, as well as for the rational development and utilization of the water, ice, and snow resources in cold regions [13,14].

Different methods have been proposed for the classification of cold regions. Koppen et al. [15] was the first to propose a division index for cold regions. Two indices were used

to classify the cold regions in Canada: (1) The average temperature of the coldest month is $\leq -3.0$ °C; and (2) The number of months with an average monthly temperature of >10 °C is less than four. Gerdel et al. [16] suggested dividing the Canadian cold regions on the basis of the criterion of an annual average temperature of 0 °C and lower. However, Wilson et al. [17] reported that there were some problems with only considering the temperature factor and proposed the use of both temperature and precipitation for cold region division. Hamelin et al. [18] proposed 10 indicators to divide the Canadian cold regions. For the classification of the cold regions in China, Yang et al. [19] proposed climate indicators on the basis of the above studies. The indicators included: (1) The average temperature in the coldest month is below −3 °C; (2) The number of months with an average monthly temperature of above 10 °C is less than four; (3) The freezing period of rivers and lakes is more than 100 days, and more than 50% of the precipitation is solid precipitation; (4) The number of months with an average monthly temperature of >10 °C is no greater than five; (5) The average temperature in October and April is below 0 °C; (6) The annual average temperature does not exceed 5 °C; (7) The number of days with an average daily temperature of >10 °C is less than 150; (8) The accumulated temperature is 500–1000 °C; (9) The percentage of solid precipitation is greater than 30%; and (10) The average annual number of snow cover days is >30 days. They divided the cold regions in China on the basis of these 10 indicators [3]. Their results show that China's cold regions were mainly distributed in four main regions: (1) Gansu, Qinghai, and Xinjiang; (2) Tibet, Aba, and Ganzi, in western Sichuan, northern Yunnan, the Yulong Mountains, and the north part of the Gaoligong Mountains in Yunnan; (3) Northeast and northwest Heilongjiang; and (4) The northeastern part of Inner Mongolia, except for the desert areas in the Junggar Basin, the Tarim Basin, and the northern part of Heihe [3,19]. The data reveals that the cold regions accounted for about 43% of the land area of China. To consolidate the various indicators proposed by Yang et al. [19], Chen et al. [20] proposed three indicators: the average temperature in the coldest month is <−3.0 °C; the number of months with an average temperature of >10 °C is less than five; and the annual average temperature is $\leq 5$ °C. Then, using these three indicators, Chen et al. [20] created a spatial distribution map of the cold regions in China from 1961 to 1998 based on observation data collected four times a day at 571 stations in China from 1961 to 1998. They reported that China's cold regions were mainly distributed in the Greater Khingan Mountains, in the Changbai Mountains, on the Sanjiang Plain in northeastern China, in the Hexi Corridor, in most of the mountainous areas in Xinjiang, and on the Qinghai-Tibetan Plateau. The cold regions covered an area of $417.4 \times 10^4$ km$^2$, which accounted for about 43.5% of the land area. Their results provide an important basis for studying China's permafrost, glaciers, stable seasonal snow, climatic divisions, and vegetation divisions, and these results are used to this day. However, 23 years have passed since Chen's study period ended (i.e., 1998). Since the beginning of the 21st century, the average annual temperature in China has risen by about 0.63 °C [21]. Therefore, it is time to revisit China's cold region divisions and to further analyze the spatial evolution characteristics of the cold regions within the context of global warming [22–24]. Currently, such studies are lacking.

In the present study, on the basis of the three indicators proposed by Chen et al. [20] and observation data collected at 612 meteorological stations in China from 1961–2019, the cold regions in China were identified, and the spatial distribution of the cold regions was analyzed in order to provide scientific references for development in the cold regions in China, and for studying the impacts of climate change in these cold regions.

## 2. Data and Methods

### 2.1. Data Sources

1.  Temperature data: The indicators used in this study were all temperature indices. The temperature data were the monthly temperature values for 612 meteorological stations in China from January 1961 to December 2019, which were downloaded from the China Meteorological Data Network (http://data.cma.cn accessed on 21

September 2020). All the 612 stations have complete data availability and there are no gaps. The distribution of the meteorological stations is shown in Figure 1.

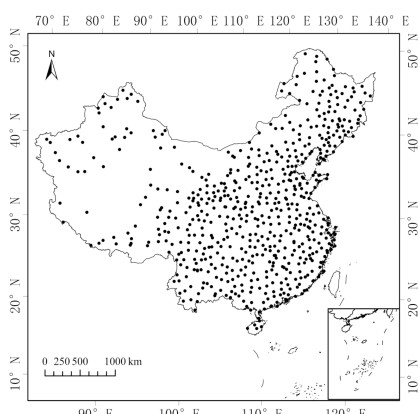

**Figure 1.** Distribution of meteorological stations in China.

2.　Elevation data: The digital elevation model (DEM) data were obtained from the cloud geospatial database (http://www.gscloud.cn/ accessed on 20 September 2020), with a resolution of 1 km × 1 km.

*2.2. Methods*

2.2.1. Spatial Interpolation of Meteorological Data

The distribution of the meteorological stations in China is uneven. For example, in remote areas such as the Qinghai-Tibetan Plateau, there are only a limited number of stations. In order to obtain high-resolution meteorological data, spatial interpolation is often used. There are many available methods of interpolating meteorological data. Of these methods, the Kriging method is commonly used. The Kriging method performs interpolation using the distribution of the meteorological elements of nearby stations, on the basis of the principle of covariance. However, since the number of meteorological stations in cold regions is small and the spatial resolution is very low, the Kriging method is not suitable. Liu et al. [25] proposed an elevation-based spatial interpolation method, i.e., ANUSPLIN. In addition to ordinary spline independent variables, this method introduces linear covariate submodels, such as the relationship between the temperature and the elevation, precipitation, and coastline. The basic principle is to allow the introduction of multivariate and covariate linear submodels. The coefficients of the models can be automatically determined on the basis of the data. This method is able to smoothly process splines of more than two dimensions, and multiple influencing factors are introduced as covariates to perform the spatial interpolation of the meteorological elements. In addition, the temperature is affected by the altitude. Because only considering the relationship between the latitude and longitude and temperature compromises the accuracy, the terrain factor was introduced in this study as a covariate. The partial thin-plate smooth spline function of the elevation linear submodel was used for the spatial temperature interpolation. The calculation formula is as follows:

$$Z_i = f(x_i) + b^T y_i + e_i (i = 1, 2, \ldots, n) \tag{1}$$

where $Z_i$ is the dependent variable at point $i$; $x_i$ is a d-dimensional spline independent variable; $f$ is the unknown smoothing function to be estimated; $y_i$ is a p-dimensional independent covariate; $b$ is the p-dimensional coefficient of $y_i$; $e_i$ is the independent variable random error with an expected value of 0 and a variance of $w_i \sigma^2$; and $w_i$ is the known local relative coefficient of variation, of which $\sigma^2$ is taken as the weight, and is the error variance, which is an unknown constant for all data points. When interpolating temperature, this

paper uses a three-variable local thin-disk smooth spline function (longitude and latitude are independent variables, and altitude is a covariate), and the number of splines is set to 2.

### 2.2.2. Trend Analysis Method

In order to reflect the change trend of the cold area from 1961 to 2019, linear trend rate estimation was adopted:

$$y = ax + b \tag{2}$$

In the formula, y is the area of the cold area; x is the year; a is the linear regression coefficient, reflecting the change trend of the area of the cold area, and a > 0 indicates that the area of the cold area is increasing, and a < 0 indicates that the area is decreasing; b is the intercept; and a $\times$ 10 is called the "climate tendency rate", and the unit is $10^4$ km$^2$/10 a.

### 2.2.3. Spatial Similarity Analysis

The kappa coefficient is generally used to determine the degree of agreement or accuracy between two images, and its calculation formula is:

$$K = \frac{P_0 - P_e}{1 - P_e} \tag{3}$$

Among them, $P_0$ is the sum of the number of samples correctly classified in each category divided by the total number of samples, which is the overall classification accuracy. Assume that the numbers of real samples in each category are $a_1$, $a_2$, ..., $a_c$, and that the predicted numbers of samples in each category are $b_1$, $b_2$, ..., $b_c$, and that the total number of samples is $n$. Then:

$$P_e = \frac{a_1 \times b_1 + a_2 \times b_2 + \ldots + a_c \times b_c}{n \times n} \tag{4}$$

The kappa coefficient calculation results are −1–1, but usually the kappa falls between 0 and 1, which can be divided into five groups to indicate the different levels of consistency: 0.0–0.20, very low consistency; 0.21–0.40, general consistency; 0.41–0.60, moderate consistency; 0.61–0.80, high consistency; and 0.81~1 are almost identical [26].

### 2.2.4. Mann–Kendall Mutation Test

In addition to the trend analysis, the MK method can also be used to test for mutation. This method is very effective for verifying a change of state from a relatively stable state to another state. For a time series, x, with $n$ sample sizes, construct an order column:

$$S_k = \sum_{i=1}^{k} r_i \ (k = 2, 3, \ldots, n) \tag{5}$$

where

$$r_i = \begin{cases} 1, \ x_i > x_j \\ 0, \ x_i \leq x_j \end{cases} \quad (j = 1, 2, \ldots, i) \tag{6}$$

It can be seen that the rank sequence, $S_k$, is the cumulative number of times the value of $i$ at the moment, $i$, is greater than the number of values at time, $j$. Under the assumption of the random independence of the time series, define statistics:

$$UF_k = \frac{[S_k - E(S_k)]}{\sqrt{Var(S_k)}} \quad (k = 1, 2, \ldots, n) \tag{7}$$

where $UF_1$ = 0; and $E(S_k)$ and $Var(S_k)$ are the mean and variance of the cumulative number, $S_k$, respectively. This value is calculated when $x_1$, $x_2$, ..., $x_n$ are independent, and when they have the same continuous distribution as:

$$E(S_k) = \frac{n(n-1)}{4} \tag{8}$$

$$Var(S_k) = \frac{n(n-1)(2n+5)}{72} \tag{9}$$

$UF_i$ is a standard normal distribution, which is a sequence calculated according to a time series ($x$) order ($x_1$, $x_2$,..., $x_n$). Given a significance level, a, in comparison with the data in the known normal distribution table, and if $UF_i > Ua$, then significant changes exist in the trend. This method can also be applied to the inverse sequence of the time series, and the above procedure can be repeated by $x_n$, $x_{n-1}$, ..., $x_1$, thus making $UF_k = -UB_k$, $k = n$, $n - 1$,..., and $UB = 0$. Given the significance level, $\alpha$, the two curves of $UF_k$ and $UB_k$ and the significant horizontal line are plotted on the same graph. If the values of $UF_k$ and $UB_k$ are greater than 0, then the sequence shows an upward trend, and values below 0 indicate a downward trend. When the value exceeds the critical line, this indicates that the rising or falling trend is significant. The range beyond the critical line is defined as the time zone of mutation. If the $UF_k$ and $UB_k$ curves appear on an intersection point, and the intersection point is between the critical line, then the intersection point corresponds to the time the mutation begins. More detailed descriptions of this method are introduced in [27].

## 3. Results and Analysis

### 3.1. Time Series Changes in the Area of Cold Regions in China from 1961 to 2019

Once the data layers corresponding to the three indicators were obtained, the three layers were superimposed to extract the overlapping area, which is the spatial distribution map of the cold regions in China from 1961 to 2019. The area of the cold regions from 1961 to 2019 was calculated, and its variation with time was obtained (Figure 2a). It can be seen that the average area of the cold regions in China from 1961 to 2019 was $427.70 \times 10^4$ km$^2$. The largest area of the cold regions occurred in 1969 ($485.92 \times 10^4$ km$^2$), and the smallest area occurred in 2007 ($368.80 \times 10^4$ km$^2$). The coefficient of variation was 0.07, indicating that the interannual variation was about 7%.

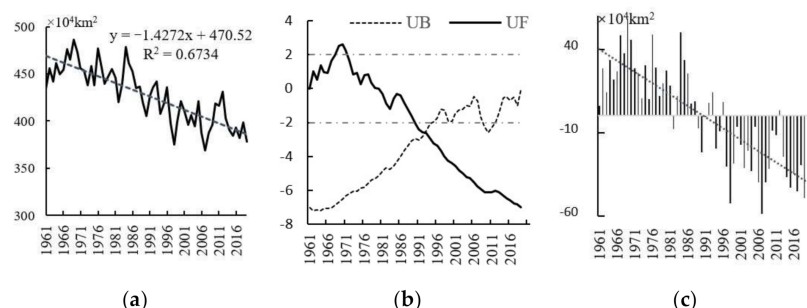

(a)　　　　　　　　　　(b)　　　　　　　　　　(c)

**Figure 2.** (**a**) Interannual changes; (**b**) MK test; and (**c**) Anomaly in the area of cold regions in China from 1961 to 2019.

The rate of change of the area of the cold regions in China from 1961 to 2019 was $-14.272 \times 10^4$ km$^2$/10 a, suggesting a significant decreasing trend ($p < 0.01$). Between 1961 and 2019, the area of the cold regions decreased by $84.20 \times 10^4$ km$^2$. The Mann–Kendall (MK) test results (Figure 2b) reveal that there have been no significant sudden changes in the area of the cold regions since 1961. However, on the basis of the area anomaly map for 1961 to 2019 (Figure 2c), the area of the cold regions in China experienced a significant turning point around 1987. From 1961 to 1987, the average area of the cold regions was $453.64 \times 10^4$ km$^2$, and from 1988 to 2019, the area decreased to $405.81 \times 10^4$ km$^2$, i.e., a reduction of $47.83 \times 10^4$ km$^2$. The results of the analysis of variance revealed that there was a significant difference in the area of the cold regions in China between 1961–1987 and 1988–2019 ($p < 0.05$). Thus, the area of the cold regions in China entered a declining stage in 1987. Furthermore, the rate of change of the area of the cold regions from 1961 to

1987 was $-5.697 \times 10^4$ km$^2$/10 a, indicating an insignificant decreasing trend; whereas the rate of change from 1988 to 2019 increased to $-11.688 \times 10^4$ km$^2$/10 a, and this change was significant ($p < 0.01$). In conclusion, the area of the cold regions in China has not only entered a relatively low value period since 1987, but it has also decreased significantly.

*3.2. Spatial Distribution of the Cold Regions in China from 1961 to 2019*

Figure 3 shows the spatial distribution of the cold regions in China from 1961 to 2019, including northeastern China, northern China, northwestern China, and southwestern China. In order to analyze the spatial distribution of China's cold regions, the cold regions were analyzed according to the administrative divisions. The cold regions were distributed in 14 provinces and autonomous regions, including Heilongjiang, Jilin, Liaoning, the Inner Mongolia Autonomous Region, Hebei, Shanxi, Shaanxi, Gansu, Ningxia, Qinghai, Sichuan, Yunnan provinces, the Xinjiang Autonomous Region, and the Tibet Autonomous Region. If you superimpose the boundaries of each administrative region on the cold area spatial distribution map calculated in this paper, using the ArcGIS Tabulate Area function, you can get the cold area of each administrative area. The areas of the cold regions in the provinces and autonomous regions are listed in Table 1, from high to low. It can be seen that the Tibet Autonomous Region contained the largest area of cold regions ($105.06 \times 10^4$ km$^2$), which was significantly larger than that of the other provinces and autonomous regions. Moreover, the areas of the cold regions in the Inner Mongolia Autonomous Region, the Xinjiang Autonomous Region, Qinghai, and Heilongjiang were also large, ranging from $54.09 \times 10^4$ km$^2$ to $75.04 \times 10^4$ km$^2$, which were significantly larger than those in Sichuan, Gansu, Jilin, and other provinces. By comparison, the cold regions in Hebei, Yunnan, Liaoning, the Ningxia Hui Autonomous Region, and Shaanxi were relatively small. Shaanxi had the smallest area of $0.32 \times 10^4$ km$^2$. The entire Tibet Autonomous Region was a cold region, except for the southern part of the Nyainqen Tanglha Mountains. In the Inner Mongolia Autonomous Region, the cold regions were mainly distributed in the northeastern part of the Inner Mongolia Plateau, and in the Yinshan, Langshan, Daqingshan, and Helan mountains. In the Xinjiang Autonomous Region, the cold regions were mainly located in the Bogda Mountains in the northern part of Hami, in the Tianshan Mountains, in the Bolokonu Mountains west of Urumqi, in the Harke Mountains in the southern part of the Tianshan Mountains, and in the Altai Mountains on the Sino-Mongolian border. All of the areas in Qinghai, except for the Qaidam Basin, were cold regions. The cold regions in Heilongjiang included the Greater Khingan Mountains, the Lesser Khingan Mountains, the Changbai Mountains, the Sanjiang Plain, and the Songnen Plain. The cold regions in Sichuan Province included the high mountains in northwestern Sichuan, such as the Daxue Mountains and the Qionglai Mountains. The cold regions in Gansu Province were mainly distributed in the Qilian Mountains in the west, in the Minshan Mountains in the Longnan area, in the northern mountains in the Hexi Corridor, in the Longzhong Plateau area, and in the Mazong Mountains in the northern part of Gansu Province. The cold regions in Jilin Province were mainly located in the Changbai Mountains in the eastern part of Jilin Province and the areas east of the Hada Mountains. The cold regions in Hebei were mainly distributed in Hengshan and Yanshan in the northwest. The cold regions in Shanxi Province were mainly distributed in the high mountains, including in the Taihang Mountains, the Lvliang Mountains, the Wutai Mountains, and the Heng Mountains. The cold regions in Yunnan Province were scattered in the southern part of the Hengduan Mountains in the northwest. The cold regions in the Ningxia Hui Autonomous Region were mainly distributed in the northern part of the Liupan Mountains. In Shaanxi, the cold regions were scattered in the middle of the Qinling Mountains.

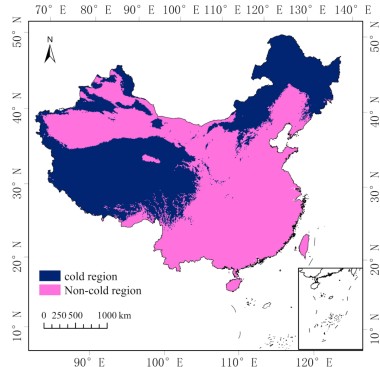

**Figure 3.** Spatial distribution of cold regions in China from 1961 to 2019.

**Table 1.** Area of cold regions in each administrative division from 1961 to 2019 ($\times 10^4$ km$^2$).

| Administrative District | Area of Cold Region | Administrative District | Area of Cold Region | Administrative District | Area of Cold Region |
|---|---|---|---|---|---|
| Tibet Autonomous Region | 105.06 | Sichuan | 19.85 | Yunnan | 1.10 |
| Inner Mongolia Autonomous Region | 75.04 | Gansu | 16.26 | Liaoning | 0.86 |
| Xinjiang Autonomous Region | 69.00 | Jilin | 11.82 | Ningxia | 0.50 |
| Qinghai | 66.82 | Hebei | 3.95 | Shaanxi | 0.32 |
| Heilongjiang | 54.09 | Shanxi | 3.03 | | |

*3.3. Spatial Variations in the Cold Regions in China from 1961 to 2019*

The World Meteorological Organization (WMO) defines the climate reference value (i.e., climatic state) as the average value of a certain meteorological element for 30 years, which is considered to be able to represent the climate of a location [28]. Thus, the 30-year average value has been used as the reference climate state in scientific research on climate and climate change. In this study, the periods of 1961–1990 and 1991–2019 were defined as two climatic states. The statistics show that the areas of the cold regions in China in 1961–1990 and 1991–2019 were 453.18 $\times 10^4$ km$^2$ and 403.86 $\times 10^4$ km$^2$, respectively. It can be seen that, against the background of global warming, the area of China's cold regions has been decreasing (i.e., by 49.32 $\times 10^4$ km$^2$). Table 2 shows the reduction in the areas of the cold regions in each administrative division. The results show that the areas of the cold regions in all 14 administrative regions decreased. Specifically, the reductions of 12.23 $\times 10^4$ km$^2$, 10.62 $\times 10^4$ km$^2$, and 8.09 $\times 10^4$ km$^2$ in Inner Mongolia, Xinjiang, and Jilin Province, respectively, were relatively large. The areas of the cold regions in Yunnan, the Ningxia Hui Autonomous Region, and Shaanxi Province decreased slightly, by 0.44 $\times 10^4$ km$^2$, 0.38 $\times 10^4$ km$^2$, and 0.27 $\times 10^4$ km$^2$, respectively.

**Table 2.** Reduction in areas of cold regions in each administrative division ($\times 10^4$ km$^2$).

| Administrative District | Area of Cold Region | Administrative District | Area of Cold Region |
|---|---|---|---|
| Inner Mongolia Autonomous Region | 12.23 | Sichuan | 1.25 |
| Xinjiang Autonomous Region | 10.62 | Liaoning | 1.21 |
| Jilin | 8.09 | Heilongjiang | 1.17 |
| Qinghai | 4.05 | Hebei | 1.12 |
| Gansu | 3.81 | Yunnan | 0.44 |
| Shanxi | 2.39 | Ningxia | 0.38 |
| Tibet Autonomous Region | 1.46 | Shaanxi | 0.27 |

On the basis of the spatial distribution maps of the cold regions in 1961–1990 (Figure 4a) and 1991–2019 (Figure 4b), the kappa values of the spatial distribution maps of the two climatic states were both 0.934, indicating excellent consistency. That is, the spatial distribution of China's cold regions did not change significantly, but there were some differences. Figure 4c shows the difference in the spatial distributions of the cold regions in China in 1991–2019 and 1961–1990. From Figure 4c, the cold regions that disappeared in the second 30-year period (1991–2019) were mainly concentrated in the central and northern parts of Jilin Province, in the central-northern parts of the Inner Mongolia Autonomous Region, in the Altai region in Xinjiang, and in the central part of the Qaidam Basin. Scattered reductions also occurred in the Hengshan, Wutai, and Yunzhong mountains in the northern part of Shanxi Province, in the southern part of the Qinling Mountains in Shaanxi Province, in the southeastern part of the Tibet Autonomous Region, in Beishan, on the Longzhong Plateau in Gansu Province, in the Qionglai Mountains and Jiajin Mountains in Sichuan Province, in the Longgang Mountains and Qianshan Mountains in Liaoning Province, in the Taihang Mountains in Hebei Province, in Southern Duanyunling in Yunnan Province, and in the northern Liupan Mountains in the Ningxia Hui Autonomous Region.

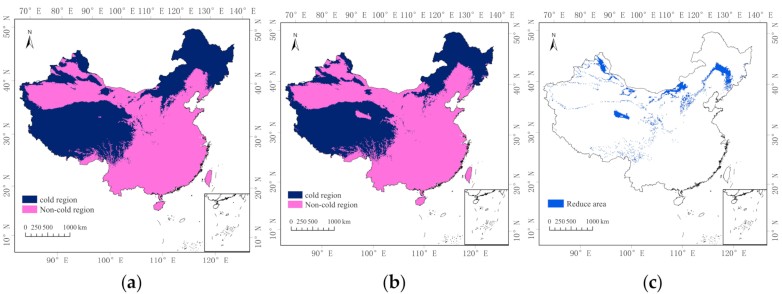

**Figure 4.** Spatial distributions of cold regions: (**a**) 1961–1990; (**b**) 1991–2019; and (**c**) Difference.

## 4. Discussion

Currently, the spatial distribution of the cold regions in China is primarily based on the results reported by Chen et al. [20]. However, they delineated the spatial distribution of China's cold regions from 1961 to 1998 without considering the variation characteristics of the cold regions. In this study, we used the observation data collected at meteorological stations in China from 1961 to 2019 to analyze the spatial distribution of China's cold regions, and we analyzed the time series changes in the area of the cold regions and the variations in their spatial distribution. The analysis of the impact of climate change on cold regions is of great scientific significance and practical value for development in the cold regions in China, and for the rational development and utilization of the water, ice, and snow resources in these cold regions.

The data period of Chen et al. [20] was from 1961 to 1998, and the data used was from 571 meteorological stations in China. In this study, the data period was from 1961 to 2019, and the data used was from 612 meteorological stations in China. In order to compare our results with those of Chen et al. [20], the cold region distribution map of Chen et al. [20] (Figure 5b) was vectorized, and the area of the distribution map was compared with that of this study (Figure 5a). The comparison results are shown in Figure 5c. The cold area obtained in this study was $427.70 \times 10^4$ km$^2$, accounting for 44.5% of the total area of China. The cold area obtained by Chen et al. [20] was $417 \times 10^4$ km$^2$, accounting for 43.5% of the total area. Thus, there was a difference of $10.3 \times 10^4$ km$^2$. As can be seen from Figure 5c, the results of this study reveal that the areas of the cold regions increased in the Qianshan and Longgang mountains in the northeastern part of Liaoning Province, in the Hengshan, Wutai, and Yunzhong mountains in Shanxi Province, in the Altay region in Xinjiang, in the Tianshan, Altun, Qilian, and Daxue Mountains, and in the Duanyun Mountains and the Mangkang Mountains on the southern Qinghai-Tibetan Plateau. In contrast, the areas of the cold regions decreased in the southern Greater Khingan Mountains, on the southern Songnen Plain, in Shanding Hural in the Inner Mongolia Autonomous Region, and in

the Qaidam Basin in Qinghai. Thus, the cold regions identified in this study are different from those identified by Chen et al. [20] because these studies were based on different research periods.

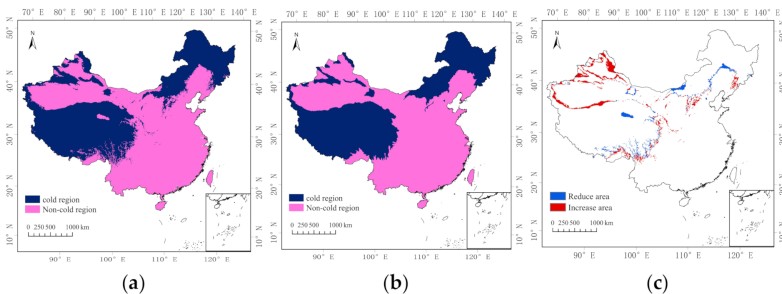

**Figure 5.** Comparison of spatial analysis results of cold regions in China: (**a**) 1961–2019; (**b**) The cold areas obtained by Chen et al.; and (**c**) Difference.

In order to compare our results with the research results of Chen Rensheng, this paper recalculated the distribution of cold regions in China from 1961 to 1998, as shown in the Figure 6 below. It can be seen that the spatial distribution map of China's cold regions calculated in this paper has higher spatial resolution than the research results of Chen Rensheng, so the statistical area should be more accurate. In addition, different spatial interpolation methods were used in the two studies, which will also make the statistical cold area different from the research results of Chen Rensheng. The main difference between the two results is due to the different time scales. More importantly, the research period of this paper is 1961–2019, which is 23 years longer than that of 1961–1998. With the extension of the research period, the spatial superposition area of the three indicators for dividing the cold region will also increase. Therefore, the area of China's cold region will increase from 1961 to 2019.

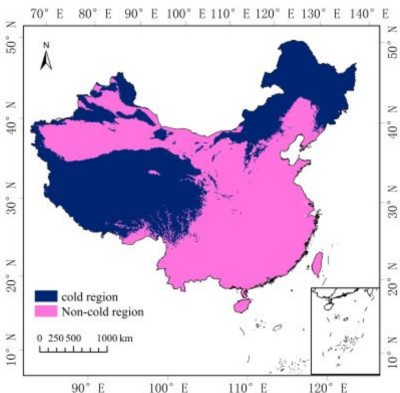

**Figure 6.** Spatial distribution of cold regions in China from 1961 to 1998.

On the basis of the area anomaly map and the results of the 5-year moving average, the year when the area anomaly became negative was 1987. Since 1987, the area of the cold regions has decreased significantly. According to the fifth assessment report of the International Panel on Climate Change (IPCC), the temperature increase became more significant after the 1980s [29]. Moreover, on the basis of the temperature data collected at 349 meteorological stations in China from 1953 to 2012, Liu et al. [30] found that the temperature increase in China during this period was significant, and that most of the years with abrupt changes were after 1986. Before the mid-1980s, the temperature in China fluctuated within a relatively small range. However, since then, the temperature has exhibited a significant upward trend. On the basis of the above discussion, it can be concluded that China's surface temperature underwent major changes in the mid-1980s.

Therefore, it is concluded that the temperature change in the mid-1980s was the main influencing factor of the change in the area of the cold regions in China.

By comparing the areas of the cold regions in China under two climatic states, it was found that the area of the cold regions in northeastern China decreased by $13.62 \times 10^4$ km$^2$, and that this was the region with the largest decrease. The results of many studies have shown that northeastern China is sensitive to climate change, and that it has experienced a significant temperature increase. For example, Chen et al. [31] studied climate change in China from 1951 to 1995 and they report that the temperature increases in China mainly occurred north of 35° N, with the largest temperature increase occurring in northern Heilongjiang. Haiying et al. [32] analyzed the spatial and temporal characteristics of the climate change in China from 1900 to 2000, and they found that the overall climate had been cooling since 1950, but that warming occurred in the northeastern, northern, and northwestern regions. The annual average temperature in northeastern China increased by about 1 °C during 1900–2000, especially from 1981 to 1998, when there was a steep jump. In addition, Liang et al. [33] analyzed the temperature characteristics in northern China from 1951 to 2014. They also found that the northeastern region had the largest rate of temperature increase, and this was also the region with the fastest temperature increase after the sudden change in temperature across the country. Therefore, the northeastern region had the largest decrease in the area of the cold regions.

The range of China's cold regions defined in this article is mainly distributed in northern Xinjiang, northeastern China, and on the Qinghai-Tibet Plateau, and this is the main distribution area of glaciers, frozen soil, and snow [34]. According to the results of this article, the area of China's cold regions is decreasing, and studies have shown that, with climate warming, glaciers shrink, the permafrost, as a whole, degenerates, snow cover shrinks, glaciers melt and shrink faster, and meltwater increases year by year. The changes in the temporal and spatial distributions of the water resources and water cycle processes caused by changes in glaciers will undoubtedly have a profound impact on the social and economic development of cold regions [35]. For example, they could lead to an increase in the glacier meltwater runoff, causing thermal melt slump, thermal melt subsidence, and other permafrost thermal melt disasters. As glacier retreat intensifies, the amount of meltwater increases, and glacier floods and glacial debris flow disasters increase with the increase in the glacier meltwater runoff [36]. Therefore, the reduction in the area of the cold region has a great impact on the ice layer, on ecosystems, and on human activities.

## 5. Conclusions

From 1961 to 2019, the area of the cold regions in China was about $427.70 \times 10^4$ km$^2$, accounting for about 44.5% of China's total land area. The rate of change in the area of the cold regions was $-14.272 \times 10^4$ km$^2$/10 a, exhibiting a very significant decreasing trend. The area of the cold regions of China decreased significantly after 1987.

The cold regions in China were mainly distributed in the Greater Khingan Mountains, in the Changbai Mountains, on the Sanjiang Plain in northeastern China, in the central part of the Inner Mongolia Plateau, and in most of the mountainous areas in Xinjiang, and on the Qinghai-Tibetan Plateau. Among the 14 provinces and autonomous regions in China, the area of the cold regions in the Tibet Autonomous Region was the largest ($105.06 \times 10^4$ km$^2$), and the area of the cold regions in Shaanxi Province was the smallest ($0.32 \times 10^4$ km$^2$).

The areas of the cold regions in China during 1961–1990 and 1991–2019 were $453.18 \times 10^4$ km$^2$ and $403.86 \times 10^4$ km$^2$, respectively. The difference between the two climatic states was $49.32 \times 10^4$ km$^2$. The largest decrease in the area of the cold regions occurred in the Inner Mongolia Autonomous Region ($12.23 \times 10^4$ km$^2$). The difference in the areas of the cold regions during the two periods was mainly distributed in northeastern China, in Xinjiang, and in Inner Mongolia.

**Author Contributions:** Y.W. analyzed the data and drafted the manuscript; L.Z., J.M., and Y.H. completed the manuscript and made major revisions; X.G. and Y.Y. completed the data visualization;

E.Z. and Y.Z. downloaded data and searched references; Y.C., N.W., and M.J. checked and proofread the manuscript. All authors have read and agreed to the published version of the manuscript.

**Funding:** This research was funded by the National Natural Science Foundation of China (Grant No. 41771067, and supported by the Key Project of Natural Science Foundation of Heilongjiang Province, No. ZD2020D002.

**Institutional Review Board Statement:** Not applicable.

**Informed Consent Statement:** Not applicable.

**Data Availability Statement:** The data presented in this study are available on request from the corresponding author.

**Conflicts of Interest:** The authors declare no conflict of interest.

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
