# Peer review of "Spatial Distribution of, and Variations in, Cold Regions in China from 1961 to 2019"

_sustainability, doi:10.3390/su14010465_

Round 1
Reviewer 1 Report
General comment
This review is on the article " Spatial Distribution of and Variations in Cold Regions in China from 1961 to 2019: The article describes the analysis of Distribution of and Variations in Cold Regions in China using station monthly temperature data.
In general, I found the article a very informative, timely requirement and it should be considered after revision. However, Authors need to follow the correct format of the journal. In addition, I found that the number of references used in this paper is much less, and the same reference is provided here and there. Need to improve the Introduction section with good citations. Especially pay more attention to the detailed comment provided in Figure 5.
I would propose a revision and will try to explain this in more detail by going through the article chapter by chapter:
Abstract
Lines 14 “it is urgent to clarify the distribution of” or it is important/emerging requirement to – think which is suitable
I believe that the finding of the research should not be represented as numbers (1) in Line 21, (2) in Line 26) and (3) in line 29.
Introduction
Lines 26-28, it is confusing to me. The rate of change means is it an annual decrease or total decrease from 1961 to 2019?
Lines 47-51 Citations are missing,
Line 56 “Gerdel et al. [8] suggested” the reference style is not matched with the journal style.
Line 57 the reference style is not matched with the journal style.
Please check there are many references, which are not tally with the journal style.
Ove role the total references used in the paper is much less, need to include at least another 10-15 relevant references to the statement made.
Line 87-88 “The cold regions covered an area of 417.4×104 km2, which accounted for about 43.5% of the land area”. (Important to add the reference), as you mention your finding the total cold area from 1961 to 2019 is 427.70.20×104. According to your finding, there is an increase in the cold areas. You should properly comment on why and how this is happening.
Lines 71-77 must be added the citations.
Data and Methods
Please check carefully the format used in the Journal and correct them accordingly.
Need to write more details on data including the accuracy (data availability). Are all the 612 stations have complete data availability or are there any gaps?
Figure 1: the map resolution is bad and needs to improve the quality (especially the resolution)
Results,
Figure 3 should be represented in High quality (resolution)
It is not clear how did you generate Figure 3 “Spatial distribution of the cold regions in China from 1961 to 2019”. Please provide the relevant technical details on that. Hope you have used the “ANUSPLIN” technique but reed to clearly express what is the parameter used in that technique.
It is also not very clear how authors generate the values in Table 1. Proposed to add a few lines to explain it in the same section.
It is also not very clear how authors generate the values in Table 1. Proposed to add a few lines to explain it in the same section.
Discussion,
Please check the journal format for discussion. In addition, as a reviewer, I am recommended not to write the discussion in separate points form.
Lines 315-317, please state what is the usefulness of having 612 stations compared to 571 stations. In addition, the Authors need to explain what the improvement you get through it is. Authors can describe based on the geographical distribution of stations as well. If the author can represent new stations in Figure 5(c) it would be helpful for interpretation. Authors can explain why the differences have occurred very easily.
Need to improve the Quality of Figure 5 (High resolution and you can make it a bit bigger for better visualization). Additionally, no need to add topographical grids on this figure as you already add them in the study area map.
Figure 5 caption also can represent the number of stations used for analysis.
Figure 5(b) legend is not visible, should keep them clear.
Figure 5(b) looks like a screenshot of an image, if it is so I am having a really doubt on how to prepare the Figure 5(c). My argument is without a raster of Figure 5 (b) author cannot produce Figure 5(c). Please prepare a good map for Figure 5(b). This is a major thing to be addressed by authors.
Conclusion
Please check the journal format for the Conclusion. In addition, as a reviewer, I am recommended not to write the discussion in separate points form.
Author Response
Thanks for your comments, my response is in the attachment.

Reviewer 2 Report
The cold zone division index is used to divide China's cold zone from 1961 to 2019, and the variations and changes in the two climatic cold zones are analyzed. Cold regions are very sensitive to climate change. There is currently no research on the changes in cold regions across China. Therefore, this research is very important and highly innovative. I would recommend the manuscript for revision.
- Methods: How accurate is the interpolation? Has the accuracy been verified? Whether the quality control of the station data is carried out before interpolation in order to minimize random and systematic errors?
- Lines 205-207: how to calculate the area anomaly, based on the average during 1961-2019 or the 30-yr climate normal periods? I think the 30-yr baseline can better reflect the relative changes in the area of cold region.
- The cold regions in China mainly located in the northern Xinjiang, northeastern China, and the Tibetan Plateau, which is the main distribution of cryosphere. Glacier and snow melt are the main sources of water supply in these regions. Therefore, I suggest that the impacts of decrease in the cold region on the cryosphere (such as permafrost, snow cover, and glacier), ecosystems and human activities should be discussed in the Discussion section.
Author Response

(The authors gave the same response as above.)

Round 2
Reviewer 1 Report
All the corrections have been made carefully and correctly by the authors and, I found the article now in good quality and it should be considered for publication.